# A Unified Understanding of Adversarial Vulnerability Regarding Unimodal Models and Vision-Language Pre-training Models

Haonan Zheng
zhenghaonan@mail.nwpu.edu.cn
Northwestern Polytechnical University
School of Electronics and Information
Xi'an, Shaanxi, China

Xinyang Deng
xinyang.deng@nwpu.edu.cn
Northwestern Polytechnical University
School of Electronics and Information
Xi'an, Shaanxi, China

Wen Jiang*
jiangwen@nwpu.edu.cn
Northwestern Polytechnical University
School of Electronics and Information
Xi'an, Shaanxi, China

Wenrui Li
liwr618@163.com
Harbin Institute of Technology
Department of Computer Science and Technology
Harbin, China

## Abstract

With Vision-Language Pre-training (VLP) models demonstrating powerful multimodal interaction capabilities, the application scenarios of neural networks are no longer confined to unimodal domains but have expanded to more complex multimodal V+L downstream tasks. The security vulnerabilities of unimodal models have been extensively examined, whereas those of VLP models remain challenging. We note that in CV models, the understanding of images comes from annotated information, while VLP models are designed to learn image representations directly from raw text. Motivated by this discrepancy, we developed the Feature Guidance Attack (FGA), a novel method that uses text representations to direct the perturbation of clean images, resulting in the generation of adversarial images. FGA is orthogonal to many advanced attack strategies in the unimodal domain, facilitating the direct application of rich research findings from the unimodal to the multimodal scenario. By appropriately introducing text attack into FGA, we construct Feature Guidance with Text Attack (FGA-T). Through the interaction of attacking two modalities, FGA-T achieves superior attack effects against VLP models. Moreover, incorporating data augmentation and momentum mechanisms significantly improves the black-box transferability of FGA-T. Our method demonstrates stable and effective attack capabilities across various datasets, downstream tasks, and both black-box and white-box settings, offering a unified baseline for exploring the robustness of VLP models.

## CCS Concepts

• **Security and privacy**; • **Information systems** → **Multimedia and multimodal retrieval**;

*Corresponding author

## Keywords

Vision-Language Models, Adversarial Attack, Transferability

**ACM Reference Format:**
Haonan Zheng, Xinyang Deng, Wen Jiang, and Wenrui Li. 2024. A Unified Understanding of Adversarial Vulnerability Regarding Unimodal Models and Vision-Language Pre-training Models. In *Proceedings of the 32nd ACM International Conference on Multimedia (MM '24), October 28-November 1, 2024, Melbourne, VIC, Australia.* ACM, New York, NY, USA, 10 pages. https://doi.org/10.1145/3664647.3681184

## 1 Introduction

ViT provides an effective Transformer-based encoder for the visual modality [11], ensuring the feature extraction of multimodal input through a unified encoding manner, significantly advancing the Vision-and-Language tasks [13, 1, 40, 45]. Various VLP models [20, 44, 38, 35] continually improve performance in V+L downstream tasks through diverse pre-training tasks and architectural designs [2, 38, 17, 24, 25]. However, the previous research in unimodal fields such as Computer Vision (CV) and Natural Language Processing (NLP) highlights the vulnerability of neural networks to adversarial attacks [12, 21]. Although adversarial robustness, particularly in CV, has been extensively explored in terms of attack strategies [4, 31], defence mechanisms [30], and transferability [9, 39], the study of adversarial robustness in VLP models remains challenges [23, 47, 48, 29]. Our study aims to develop a unified architecture to explore commonalities between multimodal and unimodal tasks from the perspective of adversarial attacks. In other words, we seek to bridge the gap, allowing rich findings in unimodal adversarial robustness to be directly applied to the multimodal scenario.

The first question we consider is "Which modality should be paid more attention?" We primarily focus on perturbations in the image modality, with perturbations in the text modality serving as orthogonal (1) **Semantic consistency**: Visual adversarial examples maintain semantic consistency, i.e., noise addition within reasonable limits doesn't change human comprehension. Conversely, text adversarial examples risk semantic distortion, potentially introducing spelling errors. (2) **Differentiability**: Image inputs are continuous and differentiable, unlike text tokens which are discrete and non-differentiable making text-only attacks less effective. (3) **Accessibility**: In real-world scenarios, text often serves as the

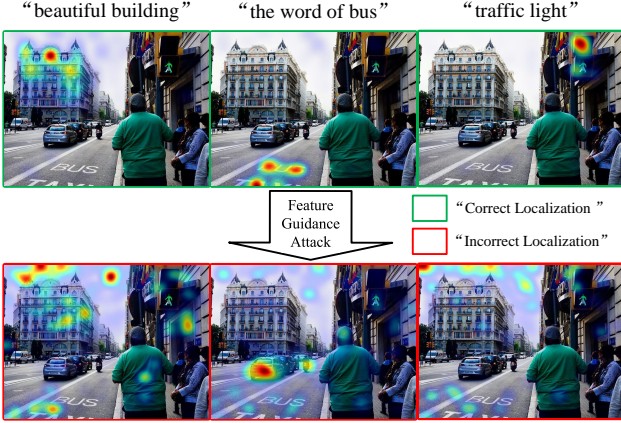

"beautiful building"  "the word of bus"  "traffic light"

Feature Guidance Attack

☐ "Correct Localization"
☐ "Incorrect Localization"

**Figure 1: ALBEF computes Grad-CAM[36] visualizations on the self-attention maps. Before FGA, ALBEF can accurately localize image content based on textual cues. After FGA, AL-BEF's understanding of the image becomes confused.**

primary means of interaction between users and AI models, with limited opportunities for attackers to modify user-generated text. In contrast, models can automatically acquire image data, simplifying the process for attackers to introduce perturbations. In fact, [29] also primarily focuses on enhancing attack strategies in the visual modality to improve adversarial transferability and [48] does not involve text attacks.

The second question is "How to unify multimodal and unimodal scenarios in exploring adversarial robustness?" We conceptualize image adversarial attacks as a feature-guided process. For unimodal models which primarily learn to understand images through detailed annotation information (such as category labels), attacking an image involves steering its embedding away from the feature vector linked to its correct annotation [12]. This deviation induces a biased comprehension of the image within the network. Alternatively, the image embedding can be guided closer to the feature vector associated with an incorrect annotation, thereby leading the network to make a predetermined error [22]. In the multimodal scenario, models are encouraged to understand images from raw text, providing a broader and more accessible source of supervision [35]. This also offers more flexible guiding information for the adversarial attack. By guiding the image embedding away from the correct text description, we induce the VLP model to develop an incorrect understanding of the image itself. Similarly, directing the embedding towards an incorrect text description intentionally misleads the model into adopting a specific erroneous interpretation. This strategy is termed Feature Guidance Attack (FGA). Expanding upon FGA, we employ adversarial texts from text attacks as guiding information to generate adversarial images, thus obtaining a novel multimodal attack. This approach exacerbates the model's misinterpretation called Feature Guidance with Text Attack (FGA-T). Furthermore, we introduce additional orthogonal mechanisms to enhance the adversarial transferability of FGA-T in the black-box scenario. Code: https://github.com/LibertazZ/FGA

Our contributions can be summarized as follows:

- We provide FGA, using original text as the supervision source for the adversarial attack on VLP models, inducing the network to misinterpret adversarial images.
- We introduce cross-modal interaction through adversarial text, forming a novel multimodal adversarial attack that enhances white-box attack strength, and improves black-box transferability through additional mechanisms.
- Our approach is theoretically orthogonal to any unimodal attack enhancement mechanism. Empirical evidence based on multiple datasets and VLP models demonstrates the broad applicability of our method to various V+L multimodal tasks, providing a unified baseline for the exploration of multimodal robustness.

## 2 Related Work

### 2.1 Unimodal Adversarial Attack

From an access perspective to the model, unimodal attacks can be divided into white-box and black-box attacks. In the black-box scenario, due to the target network's opaque weights, attackers typically conduct white-box attacks on an accessible source network, then transfer the adversarial examples to the target network. Therefore, the attack's transferability is also crucial.

**White-box Attack.** Based on how to constrain perturbation, adversarial attacks on the visual modality can generally be divided into two categories. (1) Global attacks typically involve perturbing all pixels of an image, usually constraining the distance between adversarial and original images based on $\ell_\infty$, $\ell_2$, or $\ell_1$ norms. Representative methods include FGSM [12], PGD [30], APGD [7], CW [4], etc. (2) Patch attacks, which confine the perturbation to a small area, such as 2% of the image, and allow unrestricted modification of image pixels within that area. Representative methods include LaVAN [16] and Depatch [6]. Since patch attacks are more practical, physical world attacks are usually based on this form. In the text domain, due to the discrete nature of text data, such attacks typically involve subtle modifications to the original text, such as replacing synonyms, inserting additional words, or adjusting sentence structure, without significantly altering the meaning of the text. A representative method is BertAttack [21].

**Boosting Transferability.** Enhancing the transferability of adversarial attacks is essentially a generalization problem. The two main approaches to solving the generalization issue are data augmentation and improving the optimization algorithm, thus dividing transfer attack methods into two categories. (1) Typical methods that boost transferability through data augmentation, such as DI [42] (Diverse Inputs), TI [10] (Translation Invariant) and SI [26] (Scale Invariant). (2) Typical schemes that improve optimization algorithm, such as MI [9] (Momentum Iterative), NI [26] (Nesterov Iterative), VMI [39] (Varied Momentum Iterative), VNI [39] (Varied Nesterov Iterative).

### 2.2 Multimodal Adversarial Attack

This subsection discusses relevant VLP models and multimodal adversarial attack methods.

**VLP Models.** VLP models based on different combinations of pre-training tasks can be roughly divided into three categories. (1) Aligned models: CLIP [35] contains two unimodal encoders to

align multimodal embeddings based on Image-Text Contrastive (ITC) loss. (2) Fused models by matching: ViLT [17] introduces both Image-Text Matching (ITM) and Masked Language Modeling (MLM) pre-training for V+L tasks. Models like ALBEF [20], TCL [44], BLIP [19], and VLMo [3] build on it, first aligning multimodal features using ITC loss, then fusing cross-modal features using ITM and MLM losses. (3) Fused models by Masked Data Modeling (MDM): BEiT [2] and BEiTV2 [33] propose and improve Masked Image Modeling (MIM) loss. BEiT3, based on it, first aligns multimodal features using ITC loss, then fuses cross-modal features using MIM, MLM, and Masked Language-Vision Modeling (MLVM) losses.

**Multimodal Attack.** Attacking VLP models is a novel topic. Existing work has provided valuable insights. Co-Attack [47] designs general optimization objectives based on different embeddings (unimodal or multimodal) and experimentally demonstrates that using text attacks or image attacks alone is not as effective as using both in combination, providing a general baseline for subsequent works. SGA [29] points out that improving the diversity of multimodal interaction can enhance the transferability of multimodal adversarial examples. AdvCLIP [48] provides a framework to learn a universal adversarial patch on pre-trained models for transfer attacks on downstream fine-tuned models. These works are limited to VLP models and ignore the connection between unimodal and multimodal scenarios, which is our main motivation.

## 3 Methodology

### 3.1 Feature Guidance

An image feature extractor $E$ (e.g., an image contrastive representation encoder [15, 5, 14] or a VLP model's visual encoder) projects the image into a feature vector for various visual tasks like image classification and object detection. Without regarding the subsequent usage, the most intuitive approach to generate an adversarial example $x'$ for an image $x$ is to encourage the feature vectors $E(x')$ and $E(x)$ to be as distant as possible [47]. This universal strategy is termed "Feature Deviation Attack" (FDA), which involves maximizing the loss function:

$$L_{dev} = -E(x') \cdot E(x). \tag{1}$$

where $\cdot$ represents the dot product of vectors, $E(x) \in \mathbb{R}^d$.

Assuming that in the embedding space, there exists a set of guiding vectors $W = \{\omega_i\}_{i=1}^m$, $\omega \in \mathbb{R}^d$, and there is a set of guiding labels $Y = \{y_i\}_{i=1}^n$, $y \in \{1, 2, \ldots, m\}$ specifying that $E(x')$ should be distant from the guiding vectors $\{\omega_{y_i}\}_{i=1}^n \in W$. We refer to this strategy as **"Feature Guidance Attack" (FGA)**. To realize the above concept, we need to maximize the loss function:

$$L_{gui} = -\frac{1}{n} \sum_{i=1}^n ln \left( \frac{\exp(E(x') \cdot \omega_{y_i})}{\sum_{j=1}^m \exp(E(x') \cdot \omega_j)} \right) \tag{2}$$

where $exp(\cdot)$ represents the exponential function with Euler's number $e$ as the base, and $ln(\cdot)$ stands for the logarithm to the base $e$.

Based on $L_{gui}$ or $L_{dev}$, we can apply the PGD process [30], gradually pushing the clean example $x$ along the gradient direction to maximize the loss function, ultimately obtaining the adversarial example $x'$. By the chain rule of gradients, $\frac{\partial L}{\partial x'} = \frac{\partial L}{\partial E(x')} \cdot \frac{\partial E(x')}{\partial x'}$.

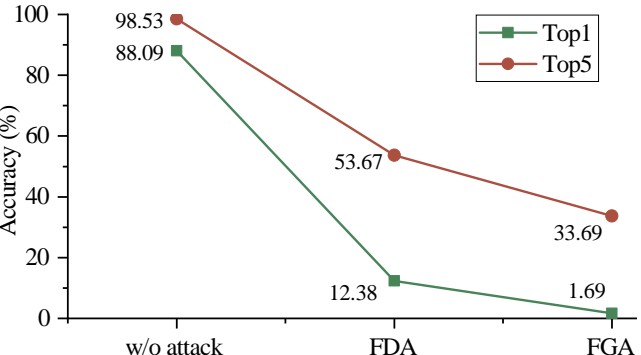

**Figure 2: Attacking results of SimCLR encoder on CIFAR-10. The reported value is classification accuracy.**

$\frac{\partial L}{\partial x'}$ represents the direction of perturbation added to the input example. While we focus on $\frac{\partial L}{\partial E(x')}$ which represents the movement direction of the feature vector:

$$\frac{\partial L_{dev}}{\partial E(x')} = -E(x) \tag{3}$$

$$\frac{\partial L_{gui}}{\partial E(x')} = \sum_{i=1}^n \left( -\frac{1}{n} \cdot \omega_{y_i} \right) + \sum_{k=1}^m \frac{exp(E(x') \cdot \omega_k)}{\sum_{j=1}^m exp(E(x') \cdot \omega_j)} \cdot \omega_k \tag{4}$$

It can be observed that feature deviation loss promotes the movement of $E(x')$ towards $-E(x)$, which means moving away from $E(x)$. While, Regarding the first term of $\partial L_{gui}/\partial E(x')$, it encourages $E(x')$ to move away from the guiding vectors $\{\omega_{y_i}\}_{i=1}^n$, and assigning equal weight $1/n$ to each of them. The second term encourages $E(x')$ to approach the guiding vector $\omega_k \in W$, with a weight of $\exp(E(x') \cdot \omega_k)/\sum_{j=1}^m \exp(E(x') \cdot \omega_j)$, which means the closer $E(x')$ is to a guiding vector, the greater the weight assigned to it. Due to the presence of the first term, $E(x')$ is far from $\{\omega_{y_i}\}_{i=1}^n$, resulting in the weight of $\omega_{y_i}$ being almost zero in the second term. Consequently, the second term effectively facilitates $E(x')$ in selecting a nearby guiding vector that does not belong to the set $\{\omega_{y_i}\}_{i=1}^n$ and moving closer to it.

We conduct a simple attack experiment using the SimCLR image encoder [5] and the CIFAR-10 dataset [18], where image feature vectors are used for image classification through a KNN-200 classifier. During the feature guidance attack, we first use the encoder to extract features for all training data. Then, by averaging the features belonging to the same category, we obtain ten guiding vectors $\{\omega_1, \omega_2, \ldots, \omega_{10}\}$. The label $y \in \{1, 2, \ldots, 10\}$ of the image $x$ serves as the guiding label, encouraging $E(x')$ to move away from the guiding vector $\omega_y$. From Table 2, it is not difficult to observe that the intensity of the feature guidance attack is greater than the feature deviation attack.

Most existing VLP models typically consist of two unimodal encoders, a text encoder $E_t$ and a visual encoder $E_v$, along with a multimodal feature fusion encoder $E_m$. For a single image-text paired example $(v, t)$, it is first mapped to a shared feature space separately by $E_t$ and $E_v$ for aligning image and text features. Subsequently, cross-modal feature fusion is conducted through $E_m$. Therefore, VLP models focus on three key embeddings: $E_v(v)$, $E_t(t)$, and $E_m(E_v(v), E_t(t))$, all corresponding to the [CLS] vector. We

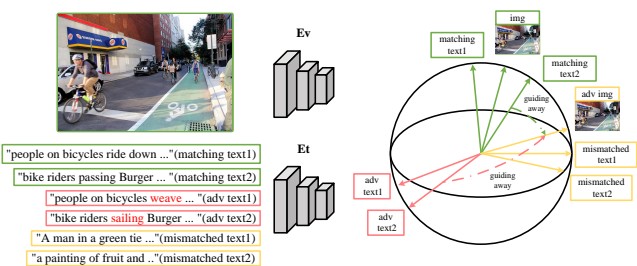

**Figure 3: Illustration of Feature Guidance with Text Attack (FGA-T) before fuse.**

will focus on finding guiding vectors in the embedding space and constructing guiding labels to execute FGA on VLP models.

## 3.2 Attacking after Fuse

In this scenario, we focus on the fused embedding $E_m(E_v(v), E_t(t))$. For different V+L downstream tasks, it needs to be fed into different subsequent models which can be uniformly understood as comprising a projector $P$ and a linear classification head $h$. $P$ projects the fused embedding into a task-specific downstream embedding space, followed by $h$ performing classification on this embedding. We can rewrite $P(E_m(E_v(v), E_t(t)))$ as $E(v|t)$, obtaining an image encoder conditioned on the textual modality. The weight of the linear classification head, $W = \{\omega_i\}_{i=1}^c \in \mathbb{R}^{d \times c}$ ($c$ is the number of categories), serve as guiding vectors. Using label information as guiding labels $Y$, we thus have all the necessary components to implement the Feature Guidance Attack. See Appendix A for more details and explanations about Visual Question Answering (VQA), Visual Reasoning (VR), etc.

FGA is primarily used to generate image adversarial examples. However, for VLP models, attacking both modalities simultaneously is a more effective strategy [47, 29]. The main challenge in generating adversarial texts involves solving the following optimization problem:

$$t' = \underset{t'}{argmax}(\|E_m(E_v(v), E_t(t')) - E_m(E_v(v), E_t(t))\|) \quad (5)$$

where BertAttack [21] is a well-suited choice for addressing this problem.

In fact, FGA and BertAttack are completely orthogonal strategies. This means we first generate an adversarial text example $t'$ and then use $E(v|t')$ as the image encoder to perform FGA, obtaining $v'$. Thus, we acquire the adversarial pair $(v', t')$.

## 3.3 Attacking before Fuse

In this scenario, only two unimodal encoders are used: $E_v$, and $E_t$. The primary intention of VLP models is to learn directly from raw text descriptions of images, utilizing a broader source of supervision [35]. This implies that in CV models, image understanding comes from pre-provided labels, such as image categories, pixel categories, or annotated bounding boxes. In contrast, in VLP models, the understanding of images originates from raw text. Consequently, using text as supervisory information to generate image adversarial examples becomes a natural approach. To implement this approach,

we first acquire a text set $T = \{t_i\}_{i=1}^m$. Then, We use the text encoder to obtain a set of guiding vectors $\{\omega_i\}_{i=1}^m = \{E_t(t_i)\}_{i=1}^m$. To obtain this text set $T$, all texts are gathered from the dataset. Here, by utilizing the dataset's annotations, we can identify which texts in the text set match with the image $v$, thereby obtaining the guiding labels $Y$. By this point, all elements necessary for executing FGA have been acquired: the image encoder $E_v$, the set of guiding vectors $\{E_t(t_i)\}_{i=1}^m$, and the guiding labels $Y$. By maximizing $L_{gui}$, the feature vector $E_v(v')$ will diverge from the text representations $\{E_t(t_y)\}_{y \in Y}$ that match $v$, thereby generating adversarial images $v'$. For more details on executing iterations of FGA and how it can be combined with typical attack strategies in the unimodal domain, refer to Appendix B.

## 3.4 Boosting Transferability before Fuse

SGA[29] points out that multimodal adversarial examples have better transferability than unimodal adversarial examples. Therefore, we need to introduce text attack into FGA before fuse. We consider an image minibatch $V = \{v_i\}_{i=1}^n$ and the text set $T_i$ represents all texts that match with the image $v_i$. Firstly, for each text $t \in T_i$, we handle the following optimization problem to generate adversarial text $t'$:

$$t' = \underset{t'}{argmax}\left(-\frac{E_t(t') \cdot E_v(v_i)}{\|E_t(t')\|_2 \|E_v(v_i)\|_2}\right) \quad (6)$$

where $\|\cdot\|_2$ denotes the Euclidean distance.

At this point, we obtain the adversarial text set $T_i'$, and we denote $T = T_1 \cup T_2 \ldots \cup T_n \cup T_1' \cup T_2' \ldots \cup T_n'$. Secondly, for $v_i \in V$, where $T_i$ is its matching texts and $T_i'$ is the adversarial texts, to generate adversarial example $v_i'$, we use the feature guidance loss to encourage $E_v(v_i')$ to simultaneously move away from both $E_t(T_i)$ and $E_t(T_i')$:

$$L_{gui}(v_i') = -\frac{1}{len_i} \sum_{t^* \in T_i \cup T_i'} log\left(\frac{\exp(E_v(v_i') \cdot E_t(t^*))}{\sum_{t \in T} \exp(E_v(v_i') \cdot E_t(t))}\right) \quad (7)$$

$len_i$ represents the length of text set $T_i \cup T_i'$.

Building on this foundation, we further introduce two strategies to enhance transferability: (1) Following SGA [29], we preset a set of resize parameters $S = \{s_1, s_2, \ldots, s_m\}$, where $h(v, s_k)$ denotes the resizing function that takes the image $v$ and the scale coefficient $s_k$ as inputs. After data augmentation, the objective function we aim to maximize is no longer $L_{gui}(v_i')$ but rather $\sum_{k=1}^m L_{gui}(h(v_i', s_k))$, where $h(v_i', s_k)$ represents the augmented image. (2) Following MI-FGSM [9], we introduce the momentum mechanism, where the current perturbation direction is determined by both the current gradient and the historical gradients from previous iterations. See Appendix B.2 for more details.

## 4 Experiments

### 4.1 Experimental Setting

*4.1.1 VLP Models.* Our experimental section involves four typical VLP models: CLIP, ALBEF, TCL and BEiT3. CLIP is a typical aligned model, consisting solely of two unimodal encoders. The latter three are fused models, containing two unimodal encoders and a multimodal encoder. ALBEF and TCL share the same architecture with some differences in the details of ITC loss. Besides, ALBEF and BEiT3 have two main differences: (1) **Different Pre-training**

**Tasks:** ALBEF is based on three pre-training tasks: ITC, ITM and MLM. In contrast, BEiT3 is based on three MDM tasks: MLM, MIM, and MVLM. (2) **Different Model Structures:** In ALBEF, the three encoders are independent of each other. BEiT3, however, uses the Multiway Transformer to split the feed-forward layer into three parallel paths, thereby obtaining three encoders.

*4.1.2 V+L Downstream Tasks.* In this part, we will introduce each downstream task involved in the experiments, along with the models and datasets used to perform these tasks.

**Visual Entailment (VE)** is a fine-grained visual reasoning task, where given a pair of $(v, t)$, the model needs to determine whether the text is supported by the image (entailment), whether the text is unrelated to the image (neutral), or whether the text contradicts the content of the image (contradictory). This task will be conducted based on the ALBEF model and the SNLI-VE [43] dataset.

**Visual Question Answering (VQA)** requires the model to predict a correct answer given an image and a question [13, 32]. It can be viewed as a multi-answer classification problem, or as an answer generation problem. We use the VQAv2 [13] dataset and the ALBEF model, which performs the VQA task through text generation.

**Visual Grounding (VG)** requires the model to find parts of the image that match the given textual description. We perform this task based on the RefCOCO+ [46] dataset and the ALBEF model.

**Visual Reasoning (VR)** requires the model to predict if a given text describes a pair of images. This task necessitates that the model not only understands the content of individual images but also compares and reasons about the relationship between two images. Therefore, the input consists of a pair of images and a piece of text. We use the BEiT3 model and the NLVR2 [37] dataset to perform this task.

**Zero-Shot Classification (ZC)** requires using predefined category descriptions (such as "a cat," "a car," etc.) as text inputs and mapping these descriptions to the embedding space by text encoder. Then, for a given image, the similarity between the image embedding and each category description embedding is calculated, and the image is classified into the category with the highest similarity. Due to the CLIP model's strong zero-shot capacity, we use it along with three datasets: CIFAR-10 [18], CIFAR-100 [18], and ImageNet [8], to perform this task.

**Image-Text Retrieval (ITR)** involves retrieving relevant images from an image database given a text query, and vice versa [45, 28, 41]. We perform this task based on the CLIP, ALBEF and TCL models, and the Flickr30k [34] and MS COCO [27] datasets.

## 4.2 Attack Effectiveness after Fuse

This subsection explores the effectiveness of attacks on the fused feature vector. Since VE, VQA, VG, and VR rely on this vector, we choose to evaluate these four tasks. As Table 1 illustrates, the evaluation follows the baseline set by [47], **TA** represents alone text attack using BertAttack. **IA** represents image attack based on feature deviation loss. **SA** stands for separate unimodal attack, indicating that TA and IA are executed separately without modal interaction, and **CA** denotes the multimodal white-box attack Co-Attack [47] which introduces cross-modal interaction. For fairness, we set the $\ell_\infty$ perturbation constraint for the image modality in FGA and FGA-T to $\epsilon = 2/255$ with 10 iterations, consistent with

**Table 1: Comparison results on four downstream tasks after fuse. The reported value is accuracy. Lower is better.**

| Method | VE | VQA | | VG | | | VR | |
|---|---|---|---|---|---|---|---|---|
| | test | dev | std | val | testA | testB | dev | test-P |
| w/o atk | 79.91 | 75.83 | 76.04 | 58.44 | 65.91 | 46.25 | 83.54 | 84.38 |
| TA | 55.09 | 45.47 | 45.89 | 49.17 | 54.05 | 39.27 | 69.59 | 70.46 |
| IA | 42.72 | 52.78 | 52.88 | 45.78 | 51.48 | 36.16 | 63.10 | 63.14 |
| SA | 38.42 | 41.21 | 41.31 | 42.13 | 45.93 | 34.96 | 58.43 | 58.53 |
| CA | 19.36 | 36.91 | 37.01 | 36.61 | 39.87 | 30.21 | 54.77 | 54.67 |
| FGA | 5.66 | 48.70 | 48.77 | 36.54 | 42.18 | 29.33 | 0.93 | 1.15 |
| FGA-T | **2.78** | **35.46** | **35.70** | **34.11** | **38.16** | **28.86** | **0.52** | **0.70** |
| FGA[1] | 39.05 | 60.66 | 60.65 | 41.68 | 45.09 | 36.41 | 27.60 | 28.79 |
| FGA-T[1] | 22.37 | 41.00 | 41.07 | 35.54 | 39.38 | 30.44 | 19.15 | 20.49 |
| FGA$_{\ell_1}$ | 8.26 | 53.47 | 53.54 | 38.70 | 44.88 | 30.76 | 1.46 | 1.74 |
| FGA$_{pat}$ | 5.23 | 51.24 | 51.22 | 55.92 | 64.23 | 45.26 | 7.59 | 8.43 |

IA, SA, and CA. Additionally, we explore the attack effectiveness when the number of iterations is 1, i.e., single-step attack, namely FGA[1] and FGA-T[1]. We also investigate the effectiveness under the $\ell_1$ constraint, namely FGAl$_{\ell_1}$, with $\epsilon_{\ell_1} = 255$ and 20 iterations. (See Appendix B.1 for more details.) Besides, we perform FGA in patch form, namely FGA$_{pat}$, with 100 iterations, a single-step $\ell_\infty$ constraint of $\alpha = 8/255$, and a patch area of 2% of the total image area, with a random location. (See Appendix B.3 for more details.) Furthermore, when involving text attack, BertAttack is used with a restriction of 1 perturbable token, following [47, 29].

We can observe from Table 1: (1) Under all tasks, FGA-T consistently achieves the best white-box attack performance, validating the effectiveness of the feature guidance approach and its orthogonality with text attack. (2) Even with only a single step, the feature guidance method is sufficient to produce effective adversarial examples, performing on par with or even better than the baseline. This provides a faster and more convenient attack strategy. (3) The feature guidance approach exhibits good orthogonality with other attack strategies in Computer Vision. When combined with $\ell_1$ attack or patch attack, it demonstrates strong performance.

## 4.3 Attack Effectiveness before Fuse

In this subsection, we explore the effectiveness of attacks on two unimodal encoders. The tasks of ZC and ITR primarily rely on two unimodal embeddings. We first conduct attacks on the ZC task. Since the text input in the ZC task is predefined and cannot be altered, we only use attacks involving the visual modality. When conducting FGA, we construct the text "A photo of a {object}." using the categorys' name to obtain the text set $T = \{t_i\}_{i=1}^c$, where $c$ represents the number of categories, following [35]. We extract features through $E_t$ to construct the guiding vectors $\{E_t(t_i)\}_{i=1}^c$, and the true category of the image serves as the guiding label $y \in \{1, 2, \ldots, c\}$. The attack results are presented in the Table 2. We observe that even FGA[1] outperforms IA which is an iterative feature deviation attack.

When executing the ITR task on the CLIP model with ViT image encoder, to construct guiding vectors for FGA, we use not only all

**Table 2: Comparison results on ZC task before fuse on CLIP. The reported value is accuracy. Lower is better.**

| Metric | Method | CIFAR-10 | CIFAR-100 | ImageNet |
|---|---|---|---|---|
| Top1 | w/o atk | 89.24 | 64.76 | 62.308 |
| | IA | 35.5 | 17.04 | 15.29 |
| | FGA | **0.01** | **0.0** | **0.004** |
| | $FGA^1$ | 30.10 | 12.21 | 5.46 |
| | $FGA_{\ell_1}$ | 14.5 | 7.06 | 0.028 |
| | $FGA_{pat}$ | 0.1 | 0.0 | 0.01 |
| Top5 | w/o atk | 98.94 | 86.9 | 86.78 |
| | IA | 78.21 | 34.89 | 31.36 |
| | FGA | 10.54 | 0.56 | **0.468** |
| | $FGA^1$ | 79.26 | 34.39 | 27.12 |
| | $FGA_{\ell_1}$ | 24.59 | 9.06 | 0.506 |
| | $FGA_{pat}$ | **9.66** | **0.35** | 0.708 |

the texts in the dataset to construct the text set ("Test Texts" in Table 3), but also follow the approach of the ZC task: using the 1000 category names from the ImageNet dataset ("ImageNet Categories" in Table 3) or the 80 category names from the MS COCO dataset's object detection task ("MS COCO Categories" in Table 3) to construct texts "There is a {object} in this photo." to form the text set $T = \{t_i\}_{i=1}^c$. Since in the Flickr30k and MS COCO datasets, an image may contain multiple objects, it is possible that the image matches multiple texts in $\{t_i\}_{i=1}^c$. In fact, we do not have annotation information indicating which objects are in the image. Therefore, we compare the cosine similarity between $E_v(v)$ and $\{E_t(t_i)\}_{i=1}^c$ to find the top 5 texts with the highest cosine similarity to $v$. When performing FGA, we encourage $E_v(v')$ to move away from the feature vectors of these five texts. From Table 3, we can summarize: (1) When using all texts to construct the feature guidance vectors, FGA achieves the best attack effect, which is intuitive. Moreover, we find that without the text attack, CLIP is already incapacitated on the ITR task. (2) ImageNet includes more categories and therefore contains richer guiding information, resulting in better attack effects compared to using categories from COCO.

Since the CLIP model only contains two unimodal encoders, attacking before fuse actually utilizes the entire CLIP model. However, the ALBEF model additionally includes a multimodal encoder, so attacking before fuse ignores the multimodal encoder. Therefore, it is necessary to validate the effectiveness of FGA before fusion on the ALBEF model. As shown in Table 4, we conduct this experiment based on the ALBEF model and the ITR task and observe phenomena consistent with Table 3.

### 4.4 Boosting Transferability

In this subsection, we transition the attack from the white-box setting to the black-box setting, which is a more common scenario. We use four VLP models: ALBEF, TCL, $CLIP_{ViT}$, and $CLIP_{CNN}$. The TCL model is identical to ALBEF except for differences in the design of the Image-Text Contrastive (ITC) loss during training, resulting in different final network weights. The two CLIP models use ViT and CNN as visual encoders, respectively. The degree of difference between these four models varies, which will inevitably

affect the transferability of adversarial examples. We will observe this phenomenon in the experiments. Our experimental setup is as follows: (1) **Task and Dataset:** We conduct black-box adversarial example transfer attacks based on the Image-Text Retrieval (ITR) task and the Flickr30k dataset. (2) **Source Model and Target Model:** The source model is the model for which we generate adversarial examples through white-box attacks, and then use them to attack the target model. Each model will serve as both source and target models. (3) **Attack Methods:** The methods we use involve attacking both image and text. SA and CA, which do not focus on transferability, serve as baselines. SGA is the state-of-the-art (SOTA) transfer attack and serves as the comparative method. FGA-$T_{aug}$ is based on FGA-T with additional data augmentation using a set of resize parameters $S$, following SGA. The differences between SGA and FGA-$T_{aug}$ are in the loss function used for generating adversarial images and the attack process (the former's attack order is "text, image, text", while the latter's attack order is "text, image"). MFGA-$T_{aug}$ additionally introduces the momentum mechanism. (4) **Hyperparameters:** All texts are allowed to modify only one word, all image perturbations are limited to 2/255 ($\ell_\infty$ norm), and the number of iterations is 10, following [47]. The resize parameters $S = \{0.5, 0.75, 1.25, 1.5\}$, following SGA.

The experimental results are shown in Table 5. We observe the following phenomena: (1) SA, CA, and SGA attack the visual modality based on feature deviation. SGA designs a more advanced set-level feature deviation and introduces data augmentation, improving both white-box and black-box attack effects on the baseline. (2) FGA-$T_{aug}$ based on feature guidance, improves SGA further, simultaneously enhancing both white-box and black-box attack effects again. (3) MFGA-$T_{aug}$ slightly reduces the white-box attack effect but further improves adversarial transferability, which is consistent with the observations in [9]. (4) Attacks based on ALBEF transfer better to TCL than to CLIP because ALBEF and TCL only have differences in parameters, while ALBEF and CLIP are completely different models. The same logic applies to attacks based on TCL. (5) Attacks based on $CLIP_{ViT}$ transfer better to $CLIP_{CNN}$ than to ALBEF or TCL because the model difference between $CLIP_{ViT}$ and $CLIP_{CNN}$ is obviously smaller than the difference with ALBEF or TCL. The same logic applies to attacks based on $CLIP_{CNN}$.

### 4.5 Visualization of Targeted Patch FGA

FGA pushes $E_v(v')$ away from matching text embeddings. Conversely, we can also push $E_v(v')$ closer to a specified text embedding to produce a predetermined error. In unimodal scenarios, this form of attack is called the targeted attack. For example, we have a text set $\{t\}_{i=1}^n$ and want to push $E(v')$ closer to a specified text $t_k$. In this case, we need to maximize the following function:

$$L_{gui}^{target} = ln\left(\frac{\exp(E(v') \cdot E_t(t_k))}{\sum_{i=1}^n \exp(E(v') \cdot E_t(t_i))}\right) \quad (8)$$

We add perturbation to the clean image $v$ in patch form, maximizing $L_{gui}^{target}$ to obtain the adversarial patch image $v'$. We execute $FGA_{patch}^{target}$ on the ALBEF model and compute Grad-CAM visualizations on the self-attention maps. As shown in Figure 4, by guiding

**Table 3: Comparison results on image-text retrieval before fuse on CLIP. For text-retrieval (TR) and image-retrieval (IR), R@1, R@5 and R@10 are reported respectively. Lower is better.**

| Method | | Flickr30k(1K test set) | | | | | | MSCOCO(5K test set) | | | | | |
| | | TR | | | IR | | | TR | | | IR | | |
| | | R@1 | R@5 | R@10 | R@1 | R@5 | R@10 | R@1 | R@5 | R@10 | R@1 | R@5 | R@10 |
| w/o attack | | 81.5 | 96.3 | 98.4 | 62.08 | 85.62 | 91.7 | 52.42 | 76.44 | 84.42 | 33.02 | 58.16 | 68.4 |
| Feature Deviation | TA | 61.8 | 85.8 | 92.0 | 41.18 | 66.68 | 76.78 | 27.42 | 51.06 | 62.54 | 16.40 | 34.49 | 44.52 |
| | IA | 25.6 | 47.6 | 56.4 | 19.84 | 39.18 | 48.94 | 10.84 | 24.16 | 32.06 | 6.76 | 17.76 | 24.71 |
| | SA | 17.2 | 35.7 | 45.9 | 11.14 | 25.32 | 33.32 | 5.8 | 14.22 | 19.8 | 3.21 | 9.39 | 14.06 |
| | CA | 7.3 | 16.5 | 22.4 | 4.18 | 10.04 | 13.86 | 1.6 | 4.62 | 7.08 | 0.94 | 2.89 | 4.53 |
| MS COCO Categories | FGA[1] | 68.9 | 89.4 | 93.5 | 49.66 | 75.08 | 83.48 | 40.46 | 64.54 | 74.08 | 23.89 | 46.57 | 57.68 |
| | FGA | 16.6 | 32.2 | 39.6 | 11.54 | 25.78 | 33.68 | 5.02 | 11.06 | 14.78 | 2.76 | 7.37 | 10.43 |
| ImageNet Categories | FGA[1] | 66.8 | 87.5 | 92.5 | 48.48 | 73.94 | 82.66 | 39.80 | 63.03 | 73.04 | 23.67 | 46.07 | 57.33 |
| | FGA | 11.5 | 21.0 | 28.3 | 7.64 | 17.8 | 23.1 | 3.54 | 8.28 | 11.44 | 2.25 | 6.02 | 8.68 |
| Test Texts | FGA[1] | 27.5 | 49.1 | 59.5 | 17.72 | 38.92 | 50.26 | 14.54 | 30.4 | 39.88 | 8.18 | 21.32 | 30.18 |
| | FGA | 0.0 | 0.8 | 1.6 | 0.14 | 0.44 | 0.96 | 0.06 | 0.24 | 0.4 | 0.024 | 0.152 | 0.264 |
| | $\text{FGA}_{\ell_1}$ | 0.1 | 0.2 | 0.5 | 0.12 | 0.32 | 0.5 | 0.04 | 0.12 | 0.16 | 0.068 | 0.200 | 0.280 |
| | $\text{FGA}_{pat}$ | 0.2 | 0.4 | 0.4 | 0.18 | 0.48 | 0.78 | 0.08 | 0.16 | 0.24 | 0.080 | 0.200 | 0.312 |

**Table 4: Comparison results on image-text retrieval before fuse on ALBEF. For text-retrieval (TR) and image-retrieval (IR), R@1, R@5 and R@10 are reported respectively. Lower is better.**

| Method | | Flickr30k(1K test set) | | | | | | MSCOCO(5K test set) | | | | | |
| | | TR | | | IR | | | TR | | | IR | | |
| | | R@1 | R@5 | R@10 | R@1 | R@5 | R@10 | R@1 | R@5 | R@10 | R@1 | R@5 | R@10 |
| w/o attack | | 95.9 | 99.8 | 100.0 | 85.5 | 97.5 | 98.9 | 77.58 | 94.26 | 97.16 | 60.67 | 84.33 | 90.51 |
| Feature Deviation | TA | 85.8 | 98.1 | 98.9 | 64.1 | 83.68 | 88.16 | 53.08 | 78.32 | 86.7 | 34.48 | 59.38 | 69.08 |
| | IA | 47.4 | 65.6 | 71.4 | 38.64 | 56.74 | 62.82 | 30.26 | 47.7 | 55.5 | 21.19 | 38.16 | 46.05 |
| | SA | 31.6 | 50.6 | 58.4 | 23.66 | 39.68 | 46.64 | 15.44 | 29.54 | 36.74 | 10.21 | 21.89 | 28.27 |
| | CA | 32.5 | 50.9 | 58.4 | 23.42 | 39.5 | 45.92 | 14.58 | 28.26 | 35.5 | 9.90 | 21.78 | 27.92 |
| Test Texts | FGA[1] | 38.0 | 58.0 | 65.3 | 31.26 | 52.12 | 60.72 | 27.96 | 48.76 | 57.88 | 20.79 | 41.26 | 51.40 |
| | FGA | **0.7** | **1.0** | **1.1** | **0.54** | **0.94** | **1.1** | **0.32** | **0.78** | **1.02** | **0.27** | **0.76** | **1.12** |

$E(v')$ closer to the prompt text through $\text{FGA}_{patch}^{target}$, ALBEF's attention area for $v'$ is concentrated on the patch, resulting in a misunderstanding.

## 4.6 FGA's Principle of Proximity

In subsection 3.1, it is mentioned that $\frac{\partial L_{gui}}{\partial E(v')}$ not only "guides $E(v')$ away from $\{\omega_{y_i}\}_{i=1}^n$", but also "selects a nearby guiding vector that does not belong to the set $\{\omega_{y_i}\}_{i=1}^n$ and moves closer to it". The performance decline of VLP models on various V+L downstream tasks in previous experiments sufficiently demonstrates the former. We further prove the latter based on the ZC task and the CLIP model. In the ZC task, we collect the text set $T = \{t_i\}_{i=1}^c$ and use it to construct the guiding vectors $\{E_t(t_i)\}_{i=1}^c$. FGA encourages $E_v(v')$ to move away from $E_t(t_y)$, where $y$ is the true category, and simultaneously encourages $E_v(v')$ to move closer to the nearest

vector from $\{E_t(t_i)\}_{i=1,i\neq y}^c$, meaning that in an ideal situation:

$$\underset{i,i\neq y}{argmax} \frac{E_v(v) \cdot E_t(t_i)}{\|E_v(v)\| \|E_t(t_i)\|} = \underset{i}{argmax} \frac{E_v(v') \cdot E_t(t_i)}{\|E_v(v')\| \|E_t(t_i)\|} \quad (9)$$

In simpler terms, the category predicted for the clean image $v$, excluding the true category $y$, will be the category predicted for the adversarial image $v'$. Based on the CIFAR-10 dataset, we present the statistical results in Figure 5. In an ideal situation, all positions except the main diagonal should be zero. We observe that the actual situation is close to the ideal. This indicates that the FGA attack indeed tends to guide "$E(v')$ to move closer to the nearest vector from $\{E_t(t_i)\}_{i=1,i\neq y}^c$". In fact, this principle of proximity promotes $v'$ to automatically choose the nearest decision boundary to cross, which is also one of the reasons for the success of FGA.

**Table 5: Compare the transferability with SOTA methods based on the Flickr30k dataset. The reported value is the attack success rate. Higher is better. R@1 value after the attack is reported in parentheses for SGA, FGA-T$_{aug}$, and MFGA-T$_{aug}$. Lower is better.**

| Source | Attack | ALBEF | | TCL | | CLIP$_{ViT}$ | | CLIP$_{CNN}$ | |
|---|---|---|---|---|---|---|---|---|---|
| | | TR R@1 | IR R@1 | TR R@1 | IR R@1 | TR R@1 | IR R@1 | TR R@1 | IR R@1 |
| ALBEF | SA | 65.69 | 73.95 | 17.60 | 32.95 | 31.17 | 45.23 | 32.82 | 45.49 |
| | CA | 77.16 | 83.86 | 15.21 | 29.49 | 23.60 | 36.48 | 25.12 | 38.89 |
| | SGA | 97.39(2.5) | 97.15(2.52) | 45.84(51.7) | 55.79(37.98) | 33.62(58.5) | 44.23(39.1) | 36.27(53.5) | 46.62(35.28) |
| | FGA-T$_{aug}$ | **99.06(0.9)** | **99.02(0.9)** | 46.89(51.2) | 58.02(35.7) | 36.07(55.9) | 47.2(36.64) | 38.95(51.0) | 50.12(32.62) |
| | MFGA-T$_{aug}$ | 97.6(2.3) | 98.15(1.64) | **52.27(45.9)** | **62.57(31.86)** | **36.93(55.4)** | **48.39(35.98)** | **39.72(50.1)** | **50.6(32.36)** |
| TCL | SA | 20.13 | 36.48 | 84.72 | 86.07 | 31.29 | 44.65 | 33.33 | 45.80 |
| | CA | 23.15 | 40.04 | 77.94 | 85.59 | 27.85 | 41.19 | 30.74 | 44.11 |
| | SGA | 49.64(48.5) | 59.85(34.78) | 98.21(1.7) | 98.79(1.1) | 34.11(57.6) | 44.68(38.64) | 37.93(52.4) | 48.47(34.02) |
| | FGA-T$_{aug}$ | 44.84(53.4) | 58.54(35.92) | **99.16(0.8)** | **99.21(0.68)** | 35.71(56.5) | 47.71(36.18) | 39.59(51.0) | 49.95(32.98) |
| | MFGA-T$_{aug}$ | **50.78(47.6)** | **63.05(32.26)** | 98.31(1.6) | 98.57(1.22) | **36.32(56.0)** | **48.94(35.36)** | **40.74(49.7)** | **50.5(32.66)** |
| CLIP$_{ViT}$ | SA | 9.59 | 23.25 | 11.38 | 25.60 | 79.75 | 86.79 | 30.78 | 39.76 |
| | CA | 10.57 | 24.33 | 11.94 | 26.69 | 93.25 | 95.86 | 32.52 | 41.82 |
| | SGA | 12.62(84.7) | 27.34(64.3) | 14.86(82.2) | 29.83(60.64) | 99.26(0.6) | 99.0(0.64) | 38.7(49.8) | 47.51(32.32) |
| | FGA-T$_{aug}$ | 12.93(84.4) | 28.84(62.7) | 14.12(82.7) | 30.12(60.44) | **99.39(0.5)** | **99.74(0.18)** | 42.78(47.3) | 48.68(31.82) |
| | MFGA-T$_{aug}$ | **13.56(83.9)** | **30.05(61.7)** | **14.96(81.8)** | **30.98(59.74)** | 99.26(0.6) | 99.52(0.36) | **44.44(46.2)** | **50.94(30.52)** |
| CLIP$_{CNN}$ | SA | 8.55 | 23.41 | 12.64 | 26.12 | 28.34 | 39.43 | 91.44 | 95.44 |
| | CA | 8.79 | 23.74 | 13.10 | 26.07 | 28.79 | 40.03 | 94.76 | 96.89 |
| | SGA | 11.16(86.1) | 25.07(66.14) | 14.12(82.6) | 27.74(62.62) | 31.17(58.8) | 42.78(37.76) | 99.74(0.2) | 99.55(0.26) |
| | FGA-T$_{aug}$ | 12.83(84.6) | 26.29(64.9) | 14.23(82.9) | 28.81(61.54) | 35.34(55.5) | 45.26(36.14) | **100.0(0.0)** | **99.93(0.04)** |
| | MFGA-T$_{aug}$ | **13.35(84.5)** | **27.48(63.88)** | **14.86(82.3)** | **30.1(60.52)** | **37.42(53.8)** | **47.2(35.04)** | 100.0(0.0) | 99.90(0.06) |

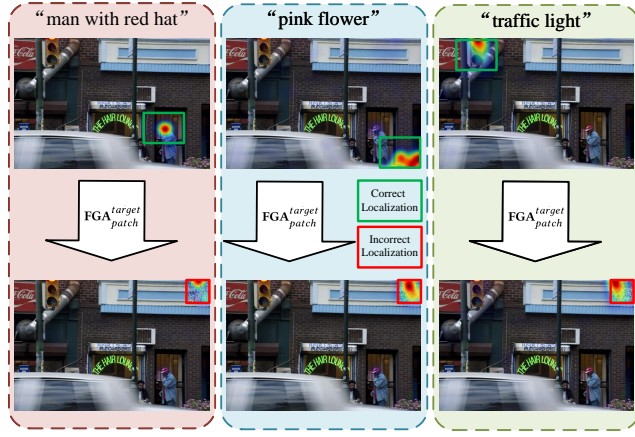

**Figure 4: Before the attack, ALBEF can accurately localize image content based on textual cues. After FGA$_{patch}^{target}$, ALBEF's attention is always erroneously focused on the patch.**

**Figure 5: Each row represents the predicted category for $v$ excluding the correct category $y$, and each column represents the predicted category for $v'$.**

## 5 CONCLUSION

In this paper, we attempt to construct a unified understanding of adversarial vulnerability regarding unimodal models and VLP models. We abstract visual modality attack into a feature guidance form and combine it with text attack and other enhancement mechanisms to establish a general baseline for exploring the security of the VLP

domain. In fact, our approach is theoretically orthogonal to many other attack schemes in the unimodal domain, which facilitates further exploration of the vulnerabilities of VLP models and the design of defence algorithms in subsequent work. We hope our code can be beneficial to the community.

# Acknowledgments

This work is supported by Shaanxi Key Research and Development Program (No. 2022ZDLGY03-04).

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
