# OpenReview forum: "A Unified Understanding of Adversarial Vulnerability Regarding Unimodal Models and Vision-Language Pre-training Models"
_acmmm.org/ACMMM/2024/Conference — MM2024 Oral_

### Official Review · Reviewer_Nu3V · 2024-04-28

**Rating:** 5
**Confidence:** 2

**Summary:**

The paper "A Unified Understanding of Adversarial Vulnerability Regarding Unimodal Models and Vision-Language Pre-training Models" explores the adversarial vulnerabilities of both unimodal and Vision-Language Pre-training (VLP) models. It introduces a novel adversarial attack method, the Feature Guidance Attack (FGA), which utilizes text representations to manipulate image data, creating adversarial images that effectively compromise VLP models. By integrating text attacks into FGA, named Feature Guidance with Text Attack (FGA-T), the method enhances its efficacy across various datasets and settings, showcasing significant improvements in attack transferability and effectiveness. This research aims to bridge the gap in adversarial robustness between unimodal and multimodal scenarios, providing a unified framework for future robustness explorations in complex multimodal tasks.

**Strengths:**

1. The FGA-T method, which incorporates text attacks into the FGA framework, has been shown to achieve superior attack effects against VLP models.
2. One of the critical contributions of this research is demonstrating the robust transferability of the FGA-T attacks. The method performs consistently across various datasets, downstream tasks, and both black-box and white-box scenarios.

**Limitations:**

1. While the proposed methods are intriguing, the paper would benefit from a more robust comparative analysis with existing state-of-the-art adversarial attack techniques, especially those known for their effectiveness against VLP models.
2. Include a discussion on the ethical implications of your findings. Offer guidelines or considerations for responsible use and dissemination of knowledge related to adversarial attacks.
3. The paper effectively identifies and exploits vulnerabilities in VLP models but does not discuss potential defensive mechanisms. Including such information could make the research more comprehensive and practically valuable.

**Suitability:**

3

---

### Official Review · Reviewer_tBqx · 2024-05-15

**Rating:** 5
**Confidence:** 3

**Summary:**

The authors introduced a novel method to generate adversarial images for vision-language pre-trained (VLP) models. The method utilizes labels from the dataset to construct guiding vectors, which guide adversarial images away from the original label and towards other labels. The method requires a white-box model to construct adversarial images, but the transferability of the method is better than pre-existing methods.

**Strengths:**

- The method proposed by the authors is novel and intuitive; furthermore, the experimental results indicate that the method is transferable and effective on different kinds of VLP models.
- Informative constraints, such as the FGA$^1$ and FGA$_{l_1}$, highlights the effectiveness of the method.
- Extensive discussion about why the proposed method is more effective compared to previous "more naive" methods.

**Limitations:**

- The structure of the Methodology section can be changed for better understanding. More specifically, first introducing the guiding vectors and the rational behind the loss might be more useful in conveying the method.
- On a minor note, are there any reasons that the models will focus on the top-right corner of the adversarial images in Figure 4? The same phenomenon is not observed in Figure 1, but the context of both figures should be the same. Is this specific to some subset of images?

**Suitability:**

3

---

### Official Review · Reviewer_3Jr5 · 2024-05-16

**Rating:** 5
**Confidence:** 3

**Summary:**

This paper proposes Feature Guidance Attack (FGA), a novel method using text representations to direct the generation of adversarial images, to bridge the gap between unimodal and multimodal tasks from the perspectives of adversarial robustness. Extensive experiments across various datasets, downstream tasks, and both black-box and white-box settings, demonstrate that the effectiveness of the proposed methods.

**Strengths:**

1.This paper introduces FGA, which encourages the feature vectors of adversarial examples to diverge from the guiding vectors. By doing so, text modality features can be integrated when constructing the guiding vectors.

2.The method incorporates text attack into FGA, achieving superior attack effects through the interaction of attacking two modalities.

3.The experiments are solid and the analysis are reasonable.

**Limitations:**

1.What is the motivation behind moving the feature vectors of adversarial examples closer to the guiding vectors? It seems more logical to move the feature vectors of adversarial examples as far away as possible.

2.Although introducing text attack into FGA is effective, this method lacks stealth and may be impractical for attackers, especially in scenarios where the text inputs are user-generated.

3.In the VQA task, what is the text 𝑡 in Equation 5? If it represents the answer to a multiple-choice question like “C”, how do you ensure that the generated adversarial text is also a choice like “A”, “B”, or “D”?

4.The indicative information in Figure 3 is insufficient, making the figure difficult to understand. More descriptive information should be added to the figure or its caption.

5.In Table 1, the performance of FGA in visual reasoning tasks is significantly superior compared to baseline methods. Additional explanation may be required to demonstrate the advantages of FGA in visual reasoning tasks.

**Suitability:**

3

---

### Official Review · Reviewer_n9c2 · 2024-05-25

**Rating:** 4
**Confidence:** 3

**Summary:**

This paper guides the perturbation of image features through textual features and classifies image content to conduct category-based guided attacks. Building on this, the paper implements further feature-guided attacks by adding textual attacks, achieving stronger attack effects in a white-box environment. By incorporating data augmentation and a momentum mechanism, the black-box transferability of the attacks is further enhanced.

**Strengths:**

1. This paper creates new features for attacking samples by utilizing the categories of image content, significantly increasing the perturbation level of the encoded entities within the images, rather than simply optimizing perturbations based on global textual information in comparison with global image information.
2. This method not only performs well after multiple iterations but also achieves relatively good attack effects with just one iteration.
3. The method enhances the effectiveness of black-box transfer attacks through data augmentation and momentum mechanisms.

**Limitations:**

1. The paper mentions that adding textual attacks can further enhance the attack effects, which is indeed shown in Tables 1, 2, and 5. However, in Tables 3 and 4, although the text states that CLIP loses its capability in the ITR task even without textual attacks, it does not provide the performance data when textual attacks are added.
2. The paper states that using momentum and data augmentation can improve the performance of black-box transfer attacks, but it does not provide a detailed description to explain the principles and reasons behind this performance improvement.
3.  There is a typo in the last paragraph of Section 2.1, "Boosing Transferability."

**Suitability:**

3

---

### Meta-Review · Area_Chair_tHfc · 2024-06-27

**Recommendation:** Accept (Oral)
**Confidence:** 4

**Metareview:**

The authors introduced a novel method to generate adversarial images for vision-language pre-trained (VLP) models. It utilizes labels from the dataset to enhance the attack. Although it is a while-box method, the transferability is well-studied.

All reviewers agree that the paper is a solid contribution and the rebuttal addressed the concerns by providing detailed experiments and analysis.

I recommend acceptance.